# Diffuse Optical Spectroscopy Assessment of Resting Oxygen Metabolism in the Leg Musculature

**DOI:** 10.3390/metabo11080496

**Published:** 2021-07-29

**Authors:** Scott E. Boebinger, Rowan O. Brothers, Sistania Bong, Bharat Sanders, Courtney McCracken, Lena H. Ting, Erin M. Buckley

**Affiliations:** 1Wallace H. Coulter Department of Biomedical Engineering, Georgia Institute of Technology, Emory University, Atlanta, GA 30332, USA; sboebin@emory.edu (S.E.B.); rowan.oakley.brothers@emory.edu (R.O.B.); sistaniameilina@gmail.com (S.B.); bharat.sanders@gmail.com (B.S.); lting@emory.edu (L.H.T.); 2Center for Research and Evaluation, Kaiser Permanente of Georgia, Atlanta, GA 30309, USA; courtney.e.mccracken@kp.org; 3Department of Rehabilitation Medicine, Division of Physical Therapy, Emory University, Atlanta, GA 30322, USA; 4Department of Pediatrics, School of Medicine, Emory University, Atlanta, GA 30322, USA; 5Children’s Healthcare of Atlanta, Children’s Research Scholar, Atlanta, GA 30322, USA

**Keywords:** near-infrared spectroscopy, diffuse correlation spectroscopy, muscle, oxygen metabolism, blood flow

## Abstract

We lack reliable methods to continuously assess localized, resting-state muscle activity that are comparable across individuals. Near-infrared spectroscopy (NIRS) provides a low-cost, non-invasive means to assess localized, resting-state muscle oxygen metabolism during venous or arterial occlusions (VO_2VO_ and VO_2AO_, respectively). However, this technique is not suitable for continuous monitoring, and its utility is limited to those who can tolerate occlusions. Combining NIRS with diffuse correlated spectroscopy (DCS) enables continuous measurement of an index of muscle oxygen metabolism (VO_2i_). Despite the lack of previous validation, VO_2i_ is employed as a measure of oxygen metabolism in the muscle. Here we characterized measurement repeatability and compared VO_2i_ with VO_2VO_ and VO_2AO_ in the medial gastrocnemius (MG) in 9 healthy adults. Intra-participant repeatability of VO_2i_, VO_2VO_, and VO_2AO_ were excellent. VO_2i_ was not significantly correlated with VO_2AO_ (*p* = 0.15) nor VO_2VO_ (*p* = 0.55). This lack of correlation suggests that the variability in the calibration coefficient between VO_2i_ and VO_2AO_/VO_2VO_ in the MG is substantial across participants. Thus, it is preferable to calibrate VO_2i_ prior to every monitoring session. Important future work is needed to compare VO_2i_ against gold standard modalities such as positron emission tomography or magnetic resonance imaging.

## 1. Introduction

Tonic muscle activity, defined as the baseline, resting-state level of electrical and/or metabolic muscle activity, is affected by numerous disease states, including Parkinson’s disease [1] and cerebral palsy [2]. However, reliable methods to assess localized tonic muscle activity are lacking. Currently, tonic muscle activity is typically evaluated using a subjective assessment that involves rating the resistance of the joint during passive movement of a limb [3,4]. Other indirect mechanical metrics such as the torque required to move the limb have also been used [5,6]. Direct measures of the muscle itself via magnetic resonance spectroscopy (MRS) and/or positron emission tomography (PET) can provide high spatial resolution images of muscle metabolic activity; however, these techniques often require exogenous contrast agents, are costly, and are unable to provide continuous measures [7]. Indirect calorimetry provides continuous measures of whole-body oxygen metabolism throughout the respiratory cycle [8,9]; however, this method is not localized to a muscle of interest. Similarly, blood samples provide measures of whole-limb metabolism that, while more localized than indirect calorimetry, are not highly localized [10]. Alternatively, electromyography (EMG) measures of electrical activity provide a more localized measure of muscle activity; however, EMG requires normalization to its resting level or maximum voluntary contraction level which hinders its ability to assess tonic muscle activity [1,11]. Additionally, EMG is susceptible to variations in adipose tissue thickness [12,13] and skin impedance [14] and therefore tonic muscle activity measured with EMG should not be compared across different measurement sessions nor compared between individuals [11]. Finally, near-infrared spectroscopy (NIRS) provides a low-cost, non-invasive means to assess resting-state regional muscle oxygen metabolism and blood flow via a brief vascular occlusion [15,16,17,18,19,20,21,22,23,24,25,26]. While this optical approach is relatively simple to perform and the quantitative values can be compared across individuals, it is not suitable for continuous monitoring, and its utility is limited to patients who can tolerate occlusion.

The combination of NIRS with diffuse correlation spectroscopy (DCS) offers an alternative low-cost, non-invasive optical approach to continuously (~1 Hz) estimate an index of muscle oxygen metabolism without the need for vascular occlusion [15,16,26,27,28,29,30,31,32]. In this hybrid approach, oxygen extraction fraction (OEF) measured with NIRS is combined with an index of blood flow (BFI) measured with DCS to yield an index of muscle oxygen metabolism (VO_2i_) via Fick’s law. Although VO_2i_ has been reported by several research groups [15,16,27,32], validation in the muscle has been limited to a single comparison with measures of whole limb metabolism [27].

Herein we compare continuous NIRS + DCS measures of BFI and VO_2i_ in the medial gastrocnemius (MG) muscle to NIRS-only measures of blood flow during venous occlusion and oxygen metabolism during venous and arterial occlusions. Moreover, we characterize the repeatability of each approach. We hypothesize that the DCS-measured BFI in the calf would be correlated to the NIRS-measured blood flow obtained via venous occlusion (as previously shown in the forearm [33]). Further, we hypothesized that the NIRS + DCS measured VO_2i_ would be correlated with NIRS-only measures of mean oxygen metabolism obtained during either a venous or arterial occlusion.

## 2. Results

A total of 9 participants were recruited for this study (5 female, 27 ± 6 years). One participant only completed a single arterial occlusion due to time constraints; three arterial occlusions from another participant were discarded because the slope of the difference between oxy- and deoxyhemoglobin (HbD) increased during occlusion; DCS data from another participant was discarded due to technical difficulties with the laser during acquisition; and NIRS data from a single epoch of one participant was also discarded due to issues with writing data to file. The remaining datasets were all included for analysis.

Table 1 outlines the mean ± standard deviation (SD) NIRS-measured resting-state optical properties, hemoglobin concentrations, and oxygen saturation for each participant. Cohort averaged resting-state total hemoglobin concentration (HbT_0_) was 48.1 ± 18.2 μM and oxygen saturation (StO_20_) was 55.5 ± 9.9%. Similarly, Table 2 reports the mean ± SD NIRS-measured VO_2AO_, VO_2VO_, and BF along with DCS-measured BFI and NIRS + DCS-measured VO_2i_ for each participant along with group averages. VO_2i_ averaged prior to the venous occlusion was not statistically significantly different from VO_2i_ averaged prior to the arterial occlusion; therefore, only mean ± standard deviation of VO_2i_ prior to arterial occlusion was reported. Although BF, VO_2AO_, and VO_2VO_ were measured at 4 source-detector separations, data is reported from 3.5 cm, which was consistently higher than the other separations (all *p* < 0.05). There were no statistically significant differences in any of the parameters reported in Table 1 and Table 2 between sexes. Excellent intra-participant repeatability was observed for VO_2AO_, VO_2VO_, VO_2i_, BF, HbT_0_, and StO_20_ as indicated by ICC > 0.75; BFI repeatability was rated good (ICC = 0.62, Table 3). Coefficient of variation (CV) was <0.27 for all measured parameters, with HbT_0_ and StO_20_ demonstrating the least amount of variation (mean CV = 0.06 and 0.03, respectively). ICC values for VO_2AO_, VO_2VO_, and BF were highest and CV values were lowest at the 3.5 cm source-detector separation; however, regardless of separation, all parameters had ICC > 0.49 and CV < 0.27 (Appendix A).

Results from the linear mixed effect model (LMM) where DCS-measured BFI averaged 15 s prior to venous occlusion was regressed on NIRS-measured BF (Figure 1) demonstrated a significant positive linear association between BF and BFI (slope = 5.8 × 10^−7^; *p* = 0.06) with a correlation, adjusted for repeated measures, of 0.70. Significant variation in the subject-specific slope estimates were not observed (*p* = 0.37).

Results from the LMM where VO_2VO_ was regressed on VO_2AO_ (Figure 2A) revealed that VO_2AO_ and VO_2VO_ were significantly and positively associated (slope = 0.96, *p* = 0.01) with a correlation, adjusted for repeated measures, of 0.51, even after accounting for significant between-subject variation in the estimated subject-specific slope (*p* = 0.041). Notably, as indicated in the Bland–Altman plot (Figure 2B), VO_2AO_ was generally higher than VO_2VO_, and the bias tended to increase as mean VO_2_ increased, resulting in a poor agreement between the two methods (CCC = 0.40).

Results from the LMM where VO_2i_ averaged 15 s prior to arterial occlusion was regressed on VO_2AO_ (Figure 3A) demonstrated significant between-subject variation in the estimated subject-specific slope (*p* = 0.06). After accounting for this random variation, we failed to detect a significant association between VO_2AO_ and VO_2i_ (slope = 0.93 × 10^−6^, *p* = 0.16) with a repeated measures adjusted correlation of 0.62. Similarly, results from the LMM where VO_2i_ averaged 15 s prior to venous occlusion was regressed on VO_2VO_ (Figure 3B) demonstrated a significant between-subject variation in the estimated subject-specific slope parameter (*p* = 0.09). After accounting for this random variation, we failed to detect a significant association between VO_2AO_ and VO_2i_ (slope = 0.25 × 10^−6^; *p* = 0.55; repeated measures adjusted correlation of 0.41).

## 3. Discussion

In this work we quantified resting oxygen metabolism in the medial gastrocnemius (MG) muscle using 3 different non-invasive, optical methods; two NIRS-only measures taken during a venous and arterial occlusion (VO_2VO_ and VO_2AO_, respectively), and a combination NIRS + DCS measure averaged 15 s prior to the venous or arterial occlusion (VO_2i_). Using the NIRS-only arterial occlusion method, oxygen metabolism was highest at the 3.5 cm separation, likely because the contribution from adipose tissue, which has lower metabolism than muscle, is smallest at this separation [21]. Mean oxygen metabolism in the calf at 3.5 cm was 0.031 mLO_2_/min/100 g, with a range of values extending from 0.011 to 0.052 mLO_2_/min/100 g (Table 2). These values, which were obtained while sitting with leg extended, are similar to previous work performed while supine [15,24], suggesting that resting-state metabolic rate may be independent of posture. Note, previous work did not account for the factor of ½ when calculating VO_2AO_ (Equation (4)), thus our VO_2AO_ values were half those previously reported. All measures of metabolism achieved excellent repeatability, as determined by the intraclass correlation coefficient >0.75, and average coefficients of variation were <0.27 (Table 3). ICCs were highest and CVs were lowest for the NIRS-only measures at 3.5 cm, presumably because the variable response of adipose tissue to occlusion is minimized [21]. Because the sensor was repositioned in between each epoch, this result suggests that sensor pressure and positioning have minimal influence on our resultant estimation, and this high repeatability agrees well with other publications [15,18,21,22,24].

To our knowledge, this work is the first to investigate the relationship between NIRS + DCS-measured VO_2i_ and the NIRS-measured VO_2VO_ and VO_2AO_ in the muscle. Despite the lack of previous validation, VO_2i_ is often employed as a measure of oxygen metabolism in the muscle. Typically VO_2i_ is first calibrated into physiological units using VO_2AO_ (or, in theory, VO_2VO_) [15,16]. However, we found no significant correlation between VO_2i_ and VO_2AO_ (*p* = 0.15, Figure 3A), nor between VO_2i_ and VO_2VO_, (*p* = 0.55, Figure 3B). Reasons for the lack of correlation may include inter-participant variations in the assumed fraction of venous blood volume in the interrogated tissue (γ, Equation (3)), blood hemoglobin concentration (Hgb, Equation (3)), and/or differences in depth sensitivity between VO_2i_ and VO_2AO_/VO_2VO_ estimations. VO_2i_ scales proportionally with Hgb and inversely with γ (Equation (6)); significant inter-participant variations in either Hgb or γ would lead to errors in VO_2i_ and could, in turn, weaken correlations with VO_2AO_/VO_2VO_. Alternatively, VO_2i_ and VO_2AO_/VO_2VO_ may sample different depths in the tissue. While depth sensitivity of all diffuse optical modalities scales roughly with source-detector separation, the exact depth sensitivity of each measure depends on tissue geometry, optical properties, and the optical modality (i.e., NIRS, DCS). Given that oxygen metabolism of adipose tissue is significantly less than muscle [21], and that the thickness of the adipose tissue layer in the MG can be appreciable and can vary widely between participants (ranging from roughly 0.1–2 cm [34]), differences in depth sensitivity between VO_2i_ and VO_2AO_/VO_2VO_ could lead to different fractions of signal that arises from adipose tissue versus muscle. Regardless of the reason, the correlation between VO_2i_ and VO_2AO_/VO_2VO_ shows that the variability in the calibration coefficient between VO_2i_ and VO_2AO_/VO_2VO_ is substantial across participants. Thus, it is preferrable to calibrate VO_2i_ prior to every monitoring session. Important future work is needed to compare VO_2i_/VO_2AO_/VO_2VO_ against other gold standard modalities like PET or MRI.

Of note, we also investigated the relationship between VO_2AO_ and VO_2VO_. We found VO_2VO_ was significantly correlated with VO_2AO_ (*p* = 0.01, Figure 2A), as has been observed in the forearm [19,22]. However, the agreement between the two methods was poor (CCC = 0.40), with a significant positive bias. This lack of agreement may be due to the fact that the venous occlusion was always performed prior to the arterial occlusion within each epoch. Because participants were permitted to readjust their legs between epochs, it is possible that 2 min was not enough time to return to a resting level of oxygen metabolism before the venous occlusion. Alternatively, because participants often reported the desire to move their legs to restore circulation at the end of each epoch, it is possible that metabolism dropped over the course of the epoch in response to restricted perfusion. Previous work found a similar reduction in VO_2VO_ with successive venous occlusions [22]. To avoid any potential effect of occlusion order, future work should randomize the order of venous and arterial occlusions. Nevertheless, given the moderate correlation between VO_2AO_ and VO_2VO_, these results suggest that venous and arterial occlusion measurements in the MG at these source-detector separations may not be interchangeable.

Finally, we also quantified and compared NIRS- and DCS-measured blood flow (BF and BFI, respectively). Our NIRS-measured blood flow ranged from 0.179 to 1.049 mL/min/100 g, which is lower than previously reported in the MG while supine (0.46 to 1.36 mL/min/100 g) [15]. This discrepancy is consistent with the known postural dependence of muscle perfusion [35]. While VO_2i_ did not correlate with VO_2AO_ or VO_2VO_, DCS-measured BFI was indeed correlated with NIRS-measured BF (*p* = 0.06, Figure 1). Previous work found a similar correlation between BFI averaged during the venous occlusion and BF in the forearm [33]. In the MG, we found BFI averaged both prior to and during the venous occlusion were correlated with BF, suggesting that blood flow quantified during a venous occlusion reflects the value of blood flow in a non-occluded state. Additionally, BFI had good repeatability (ICC = 0.62, mean CV = 0.25), which further supports the notion that BFI could potentially be used as a surrogate for BF to circumvent the need for venous occlusions to estimate blood flow.

## 4. Materials and Methods

Nine healthy, ambulatory participants >18 years were recruited. Subjects were excluded for participation if they had a history of lower extremity joint pain, contractures, major sensory deficits, evidence of orthopedic, muscular, or physical disability, evidence of vestibular, auditory, or proprioceptive impairment, orthostatic hypotension, and/or any neurological insult. All experiments were approved by the Emory University Institutional Review Board. All participants gave informed written consent before participating.

The experimental protocol is outlined in Figure 4. A pressure occlusion tourniquet (Zimmer ATS 2000 Tourniquet System) was affixed to the thigh of the dominant leg just above the knee. Participants were seated with their dominant leg and foot weight supported, and they were instructed to relax to limit hyperextension/enforced extension of the knee for the duration of the protocol. First, baseline measures of resting-state wavelength-dependent absorption and reduced scattering coefficients (μ_a_(λ) and μ_s_’(λ), respectively) were made with NIRS by gently holding an optical sensor over the outstretched medial gastrocnemius (MG). Measurements were repeated 3 times, repositioning slightly between each measure to account for local inhomogeneities in the underlying tissue. Next, the optical sensor was secured to the MG using a flexible rubber band. Care was taken to ensure adequate sensor contact with the skin without applying excessive pressure that could induce significant hemodynamic perturbations [36,37]. Continuous monitoring of dynamic changes in hemoglobin concentration (NIRS) and blood flow index (DCS) was performed during a 2-min baseline, a 30-s venous occlusion (VO, 90 mmHg tourniquet pressure), a 2-min recovery, a 30-s arterial occlusion (AO, 250 mmHg tourniquet pressure [15]), and a 5-min recovery period. The sensor was then removed, and participants were allowed to move their leg. This entire protocol was then repeated 5 times for a total of 5 arterial and 5 venous occlusions per measurement session. Upon completion of these 5 epochs, a final NIRS measure of resting-state μ_a_ (λ) and μ_s_’(λ) were made by again gently manually holding the sensor over the MG and repositioning 3 times.

All optical data were acquired using a customized frequency domain NIRS oximeter (Imagent, ISS, Champaign, IL, USA) and an in-house-built DCS system. The NIRS device utilized eight source wavelengths (690, 730, 750, 775, 785, 800, 825, and 830 nm) modulated at 110 MHz and four photomultiplier tube detectors with gain modulation of 110 MHz + 5 kHz to achieve heterodyne detection at 5 kHz. The DCS device used an 852 nm long-coherence-length laser source (iBeam Smart, TOPTICA Photonics, Farmington, NY, USA), two four-channel single photon counting modules (SPCMAQ4C-IO, Perkin-Elmer, Montreal, QC, Canada), and an eight-channel hardware correlator (Flex05-8ch, correlator.com, NJ, USA). NIRS and DCS data were acquired simultaneously (21 Hz NIRS, 1 Hz DCS) by placing an 842 nm short pass filter (FF01-842/SP-32-D, Semrock, Rochester, NY, USA) in front of each NIRS detector to mitigate crosstalk of the DCS source on the NIRS detectors.

The participant interface consisted of a custom-made optical sensor containing five source-detector pairs-four for NIRS (2.0, 2.5, 3.0, and 3.5 cm) and one for DCS (2.5 cm). These separations were chosen to maximize depth penetration while still maintaining adequate signal-to-noise ratio. For NIRS, we used customized 2.5 mm fiber bundles for both source and detection (50 μm multimode fibers, NA 0.66, FTTIIG23767, Fiberoptics Technology, Pomfret, CT, USA). For DCS, we used a 1-mm source fiber (FT1000EMT, NA 0.39, ThorLabs, Newton, NJ, USA) and seven single-mode detector fibers (780HP, Thorlabs, Newton, NJ, USA) bundled together at the 2.5-cm separation. The detected autocorrelation curves from these seven detectors were averaged to improved signal-to-noise ratio. All fibers were embedded in a rigid black 3D printed holder.

The data analysis pipeline is outlined in Figure 5 and described in depth in the following sections. Representative data obtained via this analysis pipeline is shown in Figure 6. Measures of participant-specific, wavelength-dependent optical properties (μ_a_(λ) and μ_s_’(λ)) were estimated from multi-distance measures of AC attenuation and phase shift using the semi-infinite solution to the photon diffusion equation [38] (Figure 5A). The measured μ_s_’(λ) were fit to an empirical power law relationship ${A\lambda }^{-b}$, where A is a scaling factor and b is the scattering power. The measured μ_a_(λ) were fit to the hemoglobin spectrum to estimate resting-state measures of oxy- and deoxy-hemoglobin concentrations (HbR and HbO, respectively), which were used to derive total hemoglobin (HbT = HbO + HbR), difference in hemoglobin (HbD = HbO − HbR), and tissue oxygen saturation (StO_2_ = HbO/HbT * 100%). Water concentration was assumed to be 75%. We also estimated the wavelength-dependent differential pathlength factor (DPF(λ)) that accounts for the increase in photon pathlength due to multiple scattering events using the following formula [39]:(1)$$\mathrm{DPF}\left(\lambda \right)=\frac{3{\mu_{s}}^{\prime }\left(\lambda \right)}{2\sqrt{3\mu_{a}\left(\lambda \right){\mu_{s}}^{\prime }\left(\lambda \right)+1}}$$

These measurements were made at baseline (3 repetitions), during the 2-min baseline prior to venous occlusion at the start of each epoch, and upon completion of the 5 epochs (3 repetitions) for a total of 11 values that were averaged to yield a mean resting-state estimate of of μ_a_(λ), μ_s_’(λ), DPF(λ), A, b, HbO, HbR, HbT, HbD, and StO_2_, denoted with subscript 0.

As outlined in Figure 5B and shown in Figure 6, continuous monitoring of dynamic changes in hemoglobin concentration with NIRS during occlusion epochs were estimated at each source-detector separation using the modified Beer–Lambert Law:(2)$$\log\left[\frac{\mathrm{AC}\left(\lambda ,r,t\right)}{{\mathrm{AC}}_{0}\left(\lambda ,r\right)}\right]=r\times \mathrm{DPF}\left(\lambda \right)\times \Delta \mu_{a}\left(\lambda ,r,t\right).$$
where AC(λ, r, t) is the AC amplitude of detected light measured at wavelength λ, source-detector separation, r, and time t; AC_0_(λ, r) is the mean AC amplitude measured at wavelength λ during a 1-min baseline at the beginning of each epoch prior to the venous occlusion; DPF(λ) is the wavelength-dependent differential pathlength factor obtained as described above (Equation (1)). Changes in hemoglobin concentrations (ΔHbO(r, t) and ΔHbR(r, t)) were derived from Δμ_a_(λ, r, t) at 690, 785, and 825 nm. Continuous measures of HbO(r, t), HbR(r, t), HbT(r, t), HbD(r, t), and StO_2_(t) were then quantified from these changes and the resting-state estimations obtained as described above (e.g., HbR(r, t) = HbR0 + ΔHbR(r,t)). Additionally, we estimated a continuous measure of oxygen extraction fraction (OEF):(3)$$\;\mathrm{OEF}\left(r,t\right)=\frac{{\mathrm{SaO}}_{2}–{\mathrm{StO}}_{2}\left(r,t\right)}{{\gamma *\mathrm{SaO}}_{2}}.$$

Here, SaO_2_ is the arterial oxygen saturation, and γ is the fraction of blood volume within the venous compartment of the tissue interrogated with our sensor [26]. We assumed a constant SaO_2_ of 100% and γ of 0.675 for all participants [16,26].

Continuous monitoring of dynamic changes in blood flow with DCS during occlusion epochs was estimated by fitting the measured intensity autocorrelation curves, g_2_(τ,t), for a blood flow index (BFI(t)) using the semi-infinite solution to the correlation diffusion equation and incorporating the measured resting-state μ_a0_ and μ_s_’_0_ extrapolated for 852 nm [38]. Fits were constrained to g_2_(τ,t) > 1.05. Data for a given detector were discarded if the detected photon count rate was less than 5 kHz.

Muscle oxygen metabolism (VO_2_) was measured with three distinct approaches (Figure 5C and Figure 6). The first approach, which we dub VO_2AO_, estimates VO_2_ (in units of mLO_2_/min/100 g) using the rate of hemoglobin deoxygenation during arterial occlusion [15,16,17,19,20,21,22,23]:(4)$$\;{\mathrm{VO}}_{2\mathrm{AO}}\left(r\right)=4\times \frac{{\mathrm{MW}}_{O_{2}}}{\rho_{O_{2}}\times \rho_{\mathrm{tissue}}}\times \frac{1}{2}\frac{\mathrm{dHbD}\left(r,t\right)}{\mathrm{dt}}.$$

Here $\frac{\mathrm{dHbD}\left(r,t\right)}{\mathrm{dt}}$ is the slope of HbD(r,t) versus time during the arterial occlusion estimated via linear regression using fitlm in MATLAB 2020b. The factor of ½ accounts for the fact that HbD represents the difference between 2 slopes [40]. The 4 accounts for the 4:1 ratio of oxygen to hemoglobin, MW_O2_ is the molecular weight of oxygen (32 g/mol), ρ_O2_ is the density of oxygen (1.429 g/L), and ρ_tissue_ is the assumed density of the muscle tissue (1.04 kg/L) [16].

The second approach, which we dub VO_2VO_, relates oxygen metabolism to the increase in deoxyhemoglobin during the venous occlusion [18,19,22,24,26]:(5)$$\;{\mathrm{VO}}_{2\mathrm{VO}}\left(r\right)=4\times \frac{{\mathrm{MW}}_{O_{2}}}{\rho_{O_{2}}\times \rho_{\mathrm{tissue}}}\times \frac{\mathrm{dHbR}\left(r,t\right)}{\mathrm{dt}}.$$

Here $\frac{\mathrm{dHbR}\left(r,t\right)}{\mathrm{dt}}$ is the slope of HbR(r,t) versus time during the venous occlusion estimated via linear regression. This approach assumes that the arterial input is fully saturated (i.e., SaO_2_ = 100%) [16] such that its contribution to the rate of change of HbR is negligible. This linear regression was applied over two separate windows. The first window extended from the start of cuff inflation to 25 s after the start of inflation to account for individuals who had an immediate hemodynamic response to the venous occlusion and thus limit potential accumulation of blood in the venous compartments [23]. The second window extended from when the cuff pressure reached 90 mmHg until the release of pressure to account for individuals who did not have an immediate hemodynamic response suggesting that the venous occlusion was not complete until the desired pressure of 90 mmHg was reached. The window with the greater slope was used to estimate VO_2VO_ for that epoch.

The third approach, which we dubbed VO_2i_, combined the oxygen extraction fraction measured by NIRS with the blood flow index measured by DCS to estimate an index of metabolism using Fick’s law [15,16,26,27]:(6)$$\;{\mathrm{VO}}_{2i}\left(r,t\right)=\mathrm{Hgb}\times \rho_{\mathrm{tissue}}\times A\times \mathrm{BFI}\left(t\right)\times \mathrm{OEF}\left(r,t\right).$$

Here Hgb is the blood hemoglobin concentration (assumed to be 14.1 g/dL) [16], ρ_tissue_ is the assumed density of the muscle tissue (1.04 kg/L) [16], and A is the amount of oxygen that can bind to hemoglobin (1.34 mLO_2_/g Hb). This approach provides a continuous measure of metabolism without the need for vascular occlusions. To compare VO_2i_(r,t) with VO_2AO_(r) and VO_2VO_(r), VO_2i_(r,t) was averaged over a 15-s period just prior to the arterial and venous occlusions, respectively.

The venous occlusion also allows us to estimate muscle blood flow (BF) using the mean rate of increase in total hemoglobin concentration during occlusion [15,16,18,24,33]. We capitalized on this additional information by comparing BF to BFI measured by DCS. We estimated BF using the following formula:(7)$$\;\mathrm{BF}\left(r\right)={\mathrm{MW}}_{\mathrm{Hb}}\times \mathrm{Hgb}\times \frac{\mathrm{dHbT}\left(r,t\right)}{\mathrm{dt}}.$$

Here MW_Hb_ is the molecular weight of hemoglobin (64.458 g/mol) [41] and $\frac{\mathrm{dHbT}\left(r,t\right)}{\mathrm{dt}}$ is the slope of HbT(r, t) versus time estimated via linear regression. Similarly to the calculation of VO_2VO_, the linear regression was applied over two separate windows and the window with the greatest slope was used to estimate BF for that epoch. To compare BF(r) with BFI, BFI(t) was averaged over a 15 s period just prior to the venous occlusion, as well as during the full occlusion.

Data were reported as mean ± standard deviation unless otherwise stated. A Wilcoxon rank sum test was employed to test for sex differences in each measured parameter. A Wilcoxon signed rank test was used to test for differences in BF, VO_2AO_, and VO_2VO_ between source-detector separations. Linear mixed effect models were used to examine linear relationships between VO_2i_ and VO_2AO_/VO_2VO_, between BFI and BF, and between VO_2AO_ and VO_2VO_. In these models, because multiple measurements were made on each subject, a subject-specific random slope was modeled in addition to a fixed effect intercept and slope. The significance of between-subject variation in the slope parameter was assessed by examining the significance of the random slope parameter at the 0.1 level of significance. This threshold was chosen based on the small sample size and the number of replicates. After adjusting for repeated measurements and between-subject variation, the fixed effect intercept and slope were examined. Further, we estimated a correlation coefficient (R) to aid in the interpretation of the strength of the linear relationship of the variables of interest in the presence of repeated measures [42]. For the relationship between VO_2AO_ and VO_2VO_, we also used Bland–Altman plots to graphically assess the agreement between the two variables [43] and Lin’s concordance correlation coefficient (CCC) to quantify this agreement. The CCC is the product of Pearson’s R and a bias correction factor that reflects the degree that the linear association between two variables differs from 45 deg through the origin. Finally, coefficient of variation (CV, defined as the ratio of the standard deviation to the mean across multiple measurements within a single participant) and an intraclass correlation coefficient (ICC) [44,45] were used to assess intra-participant repeatability of all metabolism and blood flow measures, as well as resting-state HbT and StO_2_. To estimate ICC, we used a two-way mixed effect, absolute agreement model that assumes experimenters remain fixed across epochs and treats intra-participant measurements as random samples. Typically, ICC values greater than 0.75 are classified as excellent repeatability, ICC between 0.6 and 0.74 are classified as good repeatability, between 0.4 and 0.59 are classified as fair, and less than 0.4 are classified as poor reliability [45].

## Figures and Tables

**Figure 1 metabolites-11-00496-f001:**
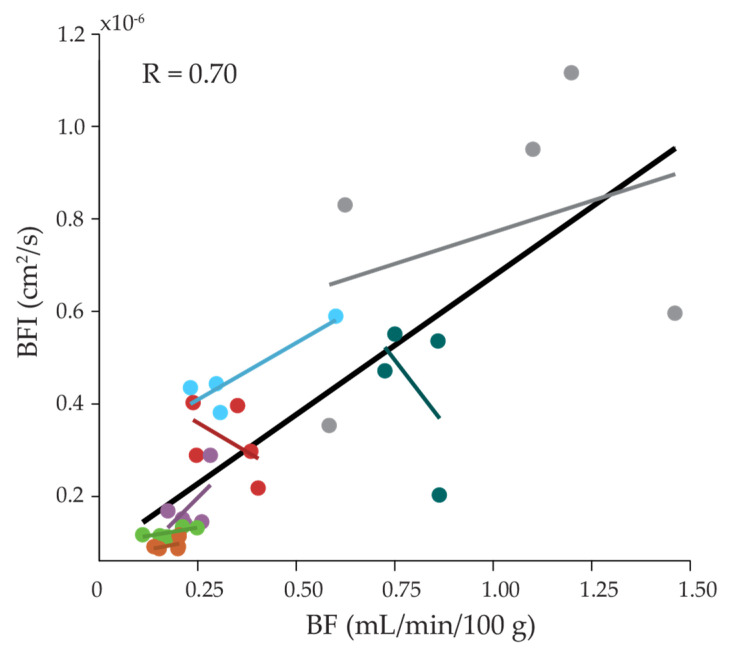
Relationship between NIRS and DCS measured blood flow. Blood flow measured during venous occlusion with NIRS (BF) at 3.5 cm versus blood flow index measured by DCS averaged 15 s prior to venous occlusion (BFI). Data are color-coded by participant. The solid black line and indicates the overall linear fit for the group data. Colored lines indicate best linear fits for individual participants.

**Figure 2 metabolites-11-00496-f002:**
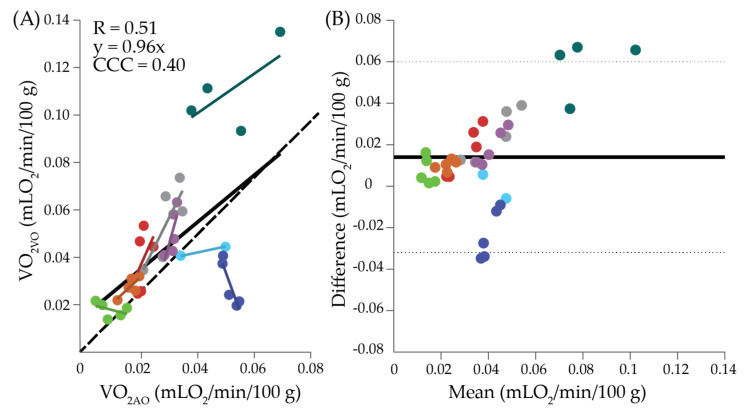
Comparison of oxygen metabolism measured with NIRS during venous and arterial occlusion. (**A**) Oxygen metabolism measured during arterial occlusion (VO_2AO_) versus oxygen metabolism measured during venous occlusion (VO_2VO_) at the 3.5-cm source-detector separation. The solid black line indicates the overall linear fit for the group data. Colored lines indicate linear fit for individual participants. The dashed line denotes the line of unity. (**B**) Bland–Altman plot of the mean of VO_2AO_ and VO_2VO_ versus the difference. The solid horizontal line indicates the mean difference, and the dotted lines indicate the 95% limits of agreement. In both figures, data are color-coded by participant.

**Figure 3 metabolites-11-00496-f003:**
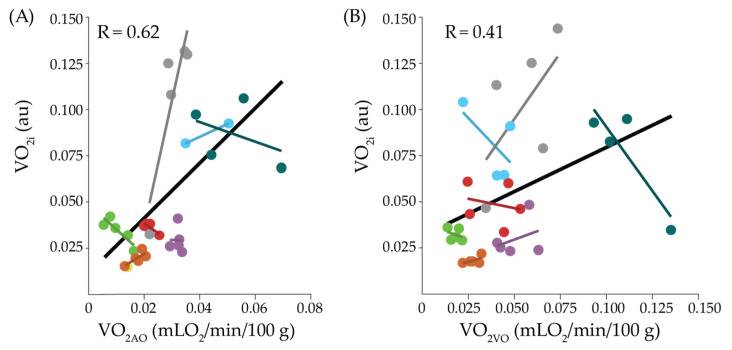
Comparison of oxygen metabolism measured with NIRS versus NIRS + DCS. (**A**) Oxygen metabolism measured with NIRS during arterial occlusion (VO_2AO_) at 3.5 cm versus NIRS + DCS-measured oxygen metabolism index (VO_2i_) averaged 15 s prior to arterial occlusion. (**B**) Oxygen metabolism measured with NIRS during venous occlusion (VO_2VO_) versus NIRS + DCS-measured oxygen metabolism index (VO_2i_), averaged 15 s prior to venous occlusion; VO_2i_ is in units of cm^2^/s*mLO_2_/g×10^-6^. The solid black line and indicates the overall linear fit for the group data. Colored lines indicate linear fit for individual participants. In both figures, data are color-coded by participant.

**Figure 4 metabolites-11-00496-f004:**
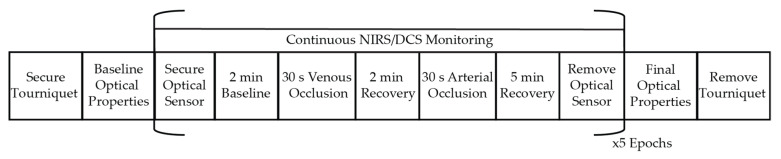
Experimental protocol. First, a tourniquet was secured to the thigh. Participants sat with their leg relaxed to limit hyperextension/enforced extension of the knee. Baseline resting-state optical properties were measured with NIRS by manually holding an optical sensor over the medial gastrocnemius (MG) muscle. Next, the optical sensor was secured to the MG for continuous NIRS/DCS monitoring throughout serial venous and arterial occlusions. This occlusion protocol was repeated 5 times, repositioning the sensor in between each epoch. Upon completion of 5 epochs, a final measure of resting-state optical properties with NIRS was made.

**Figure 5 metabolites-11-00496-f005:**
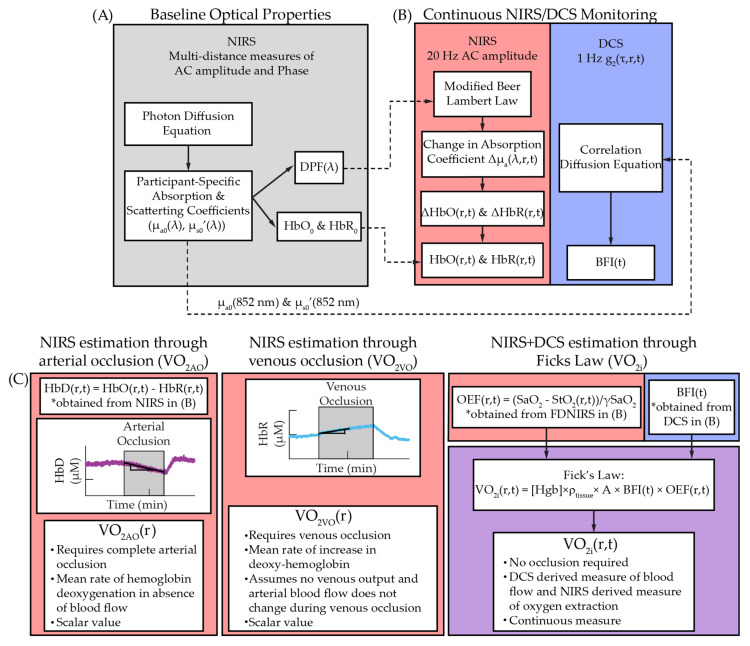
Data analysis pipeline. (**A**) First, wavelength-dependent optical properties (μ_a_(λ) and μ_s_’(λ)) were determined using the semi-infinite solution to the photon diffusion equation and were used to estimate baseline hemoglobin concentrations (HbO_0_ and HbR_0_) and the wavelength-dependent differential pathlength factor (DPF(λ)). (**B**) Next, dynamic changes in hemoglobin concentration were estimated during occlusion epochs with NIRS using the modified Beer–Lambert Law coupled with baseline variables from A (left panel, in red). Dynamic changes in blood flow index were estimated with DCS during occlusion epochs by fitting measured intensity autocorrelation curves to the semi-infinite solution of the correlation diffusion equation using baseline optical properties from A (right panel, in blue). (**C**) Finally, oxygen metabolism was calculated one of three ways: (1) using the mean rate of hemoglobin deoxygenation at each source-detector separation, r, from B during arterial occlusion (VO_2AO_(r), red box, left); (2) using the mean rate of increase in deoxyhemoglobin at each r from B during venous occlusion (VO_2VO_(r), red box, middle); (3) using Fick’s Law by combining NIRS measures of oxygen extraction from B (red box, left side) and DCS-derived measures of blood flow index from B (blue box, right side) to quantify an index of oxygen metabolism index (VO_2i_(r,t), purple box).

**Figure 6 metabolites-11-00496-f006:**
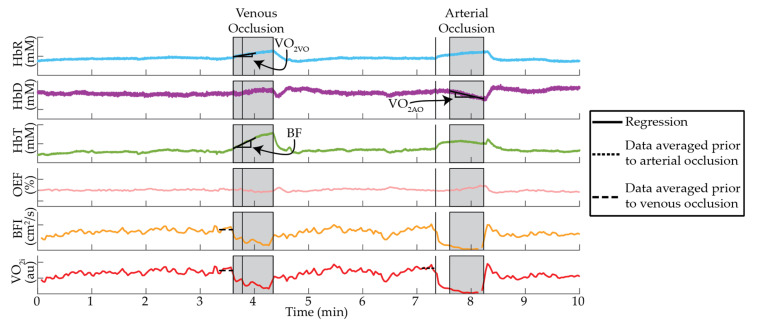
Representative cuff occlusion data. NIRS-measured deoxyhemoglobin (HbR), hemoglobin difference (HbD), total hemoglobin (HbT), and oxygen extraction fraction (OEF), along with DCS-measured blood flow index (BFI), and NIRS/DCS measured oxygen metabolism index (VO_2i_, in units of cm^2^/s*mLO_2_/g) as a function of time during a single occlusion epoch. Shaded regions indicate the venous and arterial occlusions; vertical black lines denote the start of cuff occlusion, the time point when the cuff pressure reached the desired target, and the cuff release. Solid black lines indicate the linear regressions used to quantify NIRS-only measures of blood flow and oxygen metabolism. Dotted lines indicate regions of BFI and VO_2i_ that were averaged for comparison with NIRS-only measures.

**Table 1 metabolites-11-00496-t001:** Resting-state optical properties and hemoglobin concentrations. Mean ± standard deviation of resting-state absorption and reduced scattering coefficients (μ_a0_ and μ_s0_’, respectively) at 852 nm, μ_s0_’ scaling factor (A) and scattering power (b), as well as oxy- and deoxy-hemoglobin concentrations (HbO_0_ and HbR_0_, respectively), total hemoglobin concentration (HbT_0_), and oxygen saturation (StO_20_) for each participant across 11 repetitions. Group averaged results are shown in the bottom row.

Participant	μ_a0_ (cm^−1^)	μ_s0′_ (cm^−1^)	A	b	HbO_0_ (μM)	HbR_0_ (μM)	HbT_0_ (μM)	StO_20_ (%)
1	0.163 ± 0.008	6.61 ± 0.66	485 ± 1322	0.22 ± 0.34	33.5 ± 2.9	25.7 ± 2.6	59.1 ± 3.4	56.6 ± 3.6
2	0.159 ± 0.001	5.34 ± 0.14	178 ± 130	0.41 ± 0.27	34.2 ± 0.6	22.1 ± 0.7	56.3 ± 0.4	60.8 ± 1.1
3	0.157 ± 0.004	5.71 ± 0.12	345 ± 154	0.59 ±0.08	31.57 ± 1.6	24.7 ± 1.8	56.3 ± 2.0	56.1 ± 2.4
4	0.122 ± 0.004	6.00 ± 0.40	445 ± 238	0.58 ± 0.20	26.1 ± 1.4	13.3 ± 0.5	39.4 ± 1.8	66.2 ± 0.6
5	0.146 ± 0.003	8.03 ± 0.39	134 ± 78	0.39 ± 0.10	29.9 ± 1.0	20.9 ± 0.7	50.8 ± 1.3	58.9 ± 1.1
6	0.127 ± 0.025	6.74 ± 1.18	14 ± 13	0.11 ± 0.10	25.9 ± 7.6	16.2 ± 2.8	42.1 ± 10.5	61.2 ± 2.9
7	0.205 ± 0.007	5.86 ± 0.15	216 ± 76	0.53 ± 0.05	46.5 ± 2.1	30.5 ± 1.2	77.0 ± 3.1	60.4 ± 0.8
8	0.061 ± 0.001	4.71 ± 0.06	11 ± 5	0.12 ± 0.07	4.5 ± 0.3	9.7 ± 0.3	14.2 ± 0.3	31.4 ± 1.8
9	0.109 ± 0.003	5.68 ± 0.18	21 ± 6	0.19 ± 0.04	17.9 ± 0.6	17.3 ± 1.1	35.2 ± 1.6	50.9 ± 1.2
Group	0.139 ± 0.041	6.02 ± 1.00	218 ± 469	0.37 ± 0.24	27.9 ± 1	20.3 ± 6.5	48.1 ± 18.2	55.5 ± 9.9

**Table 2 metabolites-11-00496-t002:** Resting-state oxygen metabolism and blood flow. Mean ± standard deviation oxygen metabolism measured during arterial occlusion and venous occlusion (VO_2AO_ and VO_2VO_, respectively), as well as oxygen metabolism index averaged prior to the arterial occlusion (VO_2i_, in units of cm^2^/s × mL O_2_/g × 10^−8^), blood flow measured during venous occlusion (BF), and blood flow index (BFI, in units of cm^2^/s × 10^−7^) across 5 repetitions. NA denotes data not available. Participant 6 only completed 1 arterial occlusion, and therefore no VO_2VO_, BF, or BFI values; participant 7 had no DCS data and therefore no VO_2i_ or BFI values. VO_2AO_, VO_2VO_, VO_2i_, and BF data are reported for the 3.5 cm source-detector separation.

Participant	VO_2AO_ (mLO_2_/min/100 g)	VO_2VO_ (mLO_2_/min/100 g)	VO_2i_ (au)	BF (mL/min/100 mL)	BFI (×10^−7^ cm^2^/s)
1	0.043 ± 0.011	0.039 ± 0.012	8.34 ± 2.15	0.36 ± 0.16	4.28 ± 1.10
2	0.022 ± 0.002	0.039 ± 0.013	3.62 ± 0.26	0.33 ± 0.08	3.22 ± 0.78
3	0.052 ± 0.014	0.113 ± 0.016	8.59 ± 1.54	0.78 ± 0.07	4.45 ± 1.41
4	0.030 ± 0.006	0.057 ± 0.014	10.52 ± 4.18	1.05 ± 0.32	7.69 ± 3.00
5	0.032 ± 0.002	0.051 ± 0.010	2.91 ± 0.71	0.23 ± 0.04	1.80 ± 0.62
6	0.014	NA	1.44	NA	NA
7	0.052 ± 0.003	0.029 ± 0.010	NA	0.18 ± 0.07	NA
8	0.011 ± 0.005	0.018 ± 0.004	3.40 ± 0.70	0.18 ± 0.05	1.29 ± 0.01
9	0.018 ± 0.003	0.028 ± 0.004	1.95 ± 0.34	0.18 ± 0.03	0.94 ± 0.12
Group	0.031 ± 0.016	0.045 ± 0.028	5.50 ± 3.66	0.40 ± 0.34	3.37 ± 2.55

**Table 3 metabolites-11-00496-t003:** Measurement Repeatability. Intraclass correlation coefficients (ICC) and mean± standard deviation coefficients of variation (CV) for oxygen metabolism measured during arterial occlusion and venous occlusion (VO_2AO_ and VO_2VO_, respectively), as well as oxygen metabolism index (VO_2i_), blood flow measured during venous occlusion (BF), blood flow index (BFI), total hemoglobin concentration (HbT_0_), and oxygen saturation (StO_20_).

Repeatability Measure	VO_2AO_	VO_2VO_	VO_2i_	BF	BFI	HbT_0_	StO_20_
ICC	0.88	0.85	0.77	0.83	0.62	0.99	0.99
CV	0.19 ± 0.13	0.24 ± 0.13	0.24 ±0.09	0.27 ± 0.12	0.25 ± 0.11	0.06 ± 0.07	0.03 ± 0.02

## Data Availability

The data presented in this study are available on request from the corresponding author.

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
