# Peer review of "Diffuse Optical Spectroscopy Assessment of Resting Oxygen Metabolism in the Leg Musculature"

_metabolites, 2021, doi:10.3390/metabo11080496_

Round 1

Reviewer 1 Report

Thank you to the authors for the improved version of the manuscript. I am satisfied.

Reviewer 2 Report

Dear Authors and Editors,

Although I did not find in the cover letter the precise answers for my queries from the previous review round, but I appreciate the large number of corrections that authors have performed to improve the manuscript quality. 

This manuscript is a resubmission of an earlier submission. The following is a list of the peer review reports and author responses from that submission.

Round 1

Reviewer 1 Report

Thanks for the opportunity to review this interesting manuscript. I have a few questions for the authors:
1) lines 48-50 - I am not convinced by this statement. On this basis, do the authors suggest that NIRS and spectoscopy are appropriate tools for comparative assessment than EMG?
2) I suggest that before the paragraph: results, include information about the methods and participants: I am interested in the detailed inclusion and exclusion criteria, the characteristics of the participants (age, gender, BMI), ethical information
3) If the authors examined the muscles of limb or vascular occlusion,  please specify which muscles or vascular 
4) In addition, please justify whether the amount of fat tissue may have an impact on the results or the tests procedure.

Author Response

Please see attached word document for our detailed response.

Reviewer 2 Report

Authors present their application of NIRS to time and  cost-effectively determine localized muscle activity in individuals. The study seems to address the essential need in the field for this type of robust and cheap measurements. The results displaying weak or lack of the correlation among the determined parameters carry the important information to the experts in the field. I would only ask for one additional analysis, since you have 5/4 male/female participants, could you please add a small paragraph dealing with the gender separated data and apply adequate statistics to compare. Please refer these results to the literature if there are reported, or expected gender differences, even from other species. That would potentially enhance the visibility and citations of the manuscript.

Author Response

(The authors gave the same response as above.)

Reviewer 3 Report

Dear Authors and Editors,

Here is the list of questions/comments regarding the reviewed manuscript:

  1. Abstract: PET, MRI abbreviations should be specified
  2. The main query touches the regression models which the authors summarize in the manuscript and emphasize in the abstract. In general, moderate type of regression models requires the value of determination coefficient at least 0.5. If the authors claim as the achieved result the models with R2=0.16 and 0.3, in my opinion it is not sufficient level of determination and prediction and it is not either reliable or suitable for publication in the trusted journals.  The experimantal data does not seem to fit linear trend at all, seems that the points are person-dependent as the dots seem to form something like clusters. Maybe the authors could increase the dataset and find some trends in personal data rather then searching for the universal linear fitting. Anyway, the authors should carefully recollect the data or make changes in the experimental setup or design in order to improve the quality of the models or change the idea from the search of linear trend to any other statistical approximation, for example, cluster analysis, or lots of other posibilities except simple correlations. 

Author Response

(The authors gave the same response as above.)
